# Socially Sustainable Accessibility to Goods and Services in the Metropolitan Area of Concepción, Chile, Post-COVID-19

**Francisco Núñez** [1,*], **Elías Albornoz** [1,*], **Mariella Gutiérrez** [2] and **Antonio Zumelzu** [3]

1. Laboratorio de Economía Espacial (LEE), Departamento de Planificación y Diseño Urbano, Facultad de Arquitectura Construcción y Diseño, Universidad del Bío-Bío, Concepción 4051381, Chile
2. Departamento de Ingeniería Informática, Facultad de Ingeniería, Universidad Católica de la Santísima Concepción, Concepción 4090541, Chile
3. Instituto de Arquitectura y Urbanismo, Núcleo de Investigación en Riesgos Naturales y Antropogénicos, Universidad Austral de Chile, Valdivia 5091000, Chile
* Correspondence: fnunez@ubiobio.cl (F.N.); ealborno@ubiobio.cl (E.A.)

**Abstract:** The COVID-19 pandemic affected people's mobility and access to urban activities. When the contagion was at a community level, quarantine measures were taken, causing population immobility. The lack of alternatives significantly altered the satisfaction of people's basic needs. The objective of this article was to explore and generate real accessibility indicators for goods and services, in addition to the levels of spatial satisfaction of the population, at a regional level in the metropolitan area of Concepción, Chile. To focus on citizens' social welfare, social geomarketing was applied as the method, obtaining the delimitation of accessibility areas for goods and services through population surveys and the delimited spatial decelerated satisfaction. Pre-pandemic and during-pandemic situations were evaluated. The results showed an improvement in the delimitation of accessibility areas of goods and services, as the citizens' preferences as consumers were included, revealing an increment during the pandemic, especially in the food typology. In the same way, the existence of geospatial satisfaction and its increment under the pandemic context when accessing the diverse facilities that offer these kinds of goods was confirmed. In conclusion, the satisfaction areas were useful for analyzing urban form designs and focusing them to promote revitalization, as well as for inclusive and sustainable urbanization and proactive measures to make urban areas more resilient to natural or human risks, incorporating the role of geospatial tools for promoting sustainable urban development.

**Keywords:** sustainable development; urban accessibility; urban form; social geomarketing; spatial satisfaction; cities post-COVID-19

## 1. Introduction

Human development has increased climate change, natural disasters, and socioeconomic instability. In response, the concept of sustainable development has emerged, the key elements of which are related to socioeconomic development while respecting the environment, as well as the satisfaction of needs with an adequate redistribution of resources [1]. In this sense, the concept of development must be sustained in the long term without compromising the ability of future generations to meet their own needs [2]. This is related to increasing the quality of life of people, and, in this context, sustainable development is critical for the fast-growing global population, which is estimated to reach 9700 million people by 2050 [3]. This means high demands for urban mobility in cities, as they shelter 54% of the global population. In Chile, 90% of the population lives in cities [4]. Additionally, 64% of the total movement occurs in urban spaces, and the number of total covered kilometers is expected to triple by 2050 [5]. Sustainable urban planning is conceptualized as a solution to the increase in land use [6], the extension of the urban area beyond the walkable range [7], the decrease in urban density [8], and the processes

of suburbanization [9]. In addition, the effects of these factors on the quality of life and well-being of people associated with the urban form of the territorial units they inhabit are included, such as social stress and segregation and high crime rates, among others [10–28]. Consequently, sustainable urban planning integrates aspects of the aesthetics, sustainability, and functionality of cities to provide their inhabitants or users with a higher quality of life [11,12]. Habitability in intermediate and metropolitan cities depends on sustainable mobility to generate more accessible cities and improve citizens' quality of life [13]. In this way, sustainability also covers the movement of people and can be applied to mobility and accessibility issues. Regarding this matter, solutions with the objectives of reducing the aforementioned negative externalities and increasing the satisfaction of the users of urban mobility systems can be identified [14–16]. In this context, sustainable accessibility seeks to reduce the need for motorized displacement and the maximum use of the autonomous capacity of humans to move [17]. Based on this and considering the management of sustainable accessibility, actions should be increased to increase short-distance pedestrian travel and reduce long-distance travel in private and public motorized vehicles [23].

In December 2019, in Wuhan, China, the first infection was recorded. About a month later, the World Health Organization (WHO, Geneva, Switzerland) declared the infection a public emergency of international concern, and in mid-February of the same year, it was named COVID-19. Since the appearance of the first outbreaks in different countries, respective authorities have been obliged to take precautions in order to stop the spread of the disease. These precautions include partial and total lockdown measures, which suppose a cease in a great part of the economic activity in the region where it is applied, in addition to an intrusion of the daily life of the general population, generating a health and well-being decline [18,19]. Architecture and urban spaces, which, in times of natural disasters, such as earthquakes and hurricanes, offered a good response, showed little resilience during the pandemic: the attractive model of a compact city favors dense urban areas, which act as epicenters of virus spread [18], increasing the necessity for lockdowns to counteract this problem and strengthening the crisis given the immobility of the population.

In Chile, the COVID-19 pandemic has affected people's mobility and access to urban activities. In phase 4, the contagion is communal, resulting in quarantine measures and generating population immobility due to the lack of alternatives, thereby significantly altering the satisfaction of basic needs. In Chile, the greatest reduction occurred in the use of the subway (55%), ridesourcing (51%), and buses (45%) [20]. In the main cities of the world, such as Moscow, New Delhi, Tokyo, London, Paris, and others, there were strong reductions in the road traffic volumes, with values from 54% to 11%, [21] in the year 2020 compared with the year 2019. Specifically in Poland, decreases in the range of 29.49% to 46.88% were observed in week 14 of 2020 [22]. In this emergency context and with a tendency toward urban unsustainability, access to basic goods and services intersects the visions of urban mobility. This intersection of accessibility is oriented to generating actions to improve access to public transportation, reduce commuting fares, and especially reduce distances [23]. This is related to an appropriate urban form that moves toward sustainability, in which the activities are closer to the people, to improve access and hence generate a satisfactory experience [24–28].

However, urban accessibility is usually understood only from a physical–causal perspective, which has prestige for political decisions and urban planning due to its efficacy and methods for scientific validation [29]. These records favor only an analytical contextualization from the material elements present in the geographical space, typical of transportation studies [30]. Although accessibility is traditionally evaluated through the urban form that a unit features, this indicator only considers the total available dependencies to delimit an area of pedestrian accessibility, which would only consider the availability of facilities present in the territory, extending the "local dimensions of material elements present in the geographical space". In this context, according to Lucas [31], the study of accessibility from a social perspective is still incipient at a global level, making it necessary

to advance the adoption of new methodologies that account for the current phenomena and respond to the challenges of socially sustainable mobility [32].

Based on the above, the problem of the traditional principle to understand the role of accessibility in urban planning is its limitation to analyze the socioeconomic–spatial reality concretely. The shortcoming of the approach is its validation, assuming the cost of eliminating the role of subjects and social groups in the production and transformation of the urban space from the analysis [33]. Consequently, the experiences of movement and the satisfaction declared by residents concerning access to activities related to the acquisition of basic goods and services would not be analyzed. For the purpose of favoring the social welfare of citizens, social geomarketing was applied as a supporting tool, obtaining the delimitation of the areas of accessibility to goods and services, as well the spatial satisfaction for the declared movement experience, through surveys of the population. In this form, the objective of this research was to explore the spatial satisfaction levels in the experience of access at the moment of demand for basic consumer goods and services through the evaluation of declared necessity before and during the COVID-19 pandemic. The research questions are: Is it possible to include, through indicators, the movement preferences declared by citizens in the accessibility analysis typical of urban form? Is it possible to spatially delimit the levels of satisfaction of citizens to be included in sustainable urban design?

This article presents a geospatial analysis to study, delimit, and represent accessibility indicators for goods and services, in addition to the levels of spatial satisfaction of the population in pre-established areas due to the closing of urban perimeters ruled by sanitary authorities. The study was carried out in the metropolitan area of Greater Concepción, in the Collao sector of the city of Concepción, Chile. The importance of this study lies in the generation of new strategies for the analysis of diverse current scenarios and the possibility of simulated ones in the future, which may include the installation of new infrastructure for public spaces and the identification of mobility barriers or changes in population variables, among others. In summary, this analysis is an original application that, by using social geomarketing tools and geospatial analysis, incorporates quantitative–spatial techniques for real cases. With this contribution, accessibility indicators and estimates of geospatial satisfaction pre-pandemic and during the pandemic were obtained, which are useful for decision-making in sustainable urban development and regional management post-COVID-19. Finally, this research provides a tool of interdisciplinary analysis that allows people's sustainable accessibility to and spatial satisfaction regarding basic consumer goods and services to be quantified as a new criterion of urban design of public spaces. This tool could be applicable to strategies that require support for increasing green public spaces, non-motorized means of transport, and improved participatory and inclusive processes in future, post-COVID-19 cities.

Next, a brief review of the literature is presented in Section 2. Section 3 provides the materials and methods used to answer the research questions, which allowed the methodological development of this work. Consequently, the main results obtained are presented in Section 4. The discussion of these is presented in Section 5, and the main conclusions are presented in Section 6, which show the importance of this type of analysis as a tool to support urban design decisions related to green public spaces, non-motorized means of transport, and the improvement of participatory and inclusive processes in the global south.

## 2. Literature Review

### 2.1. Accessibillity

Accessibility is a key morphological concept in urban and regional planning, given its ability to link the activities of people and companies to the possibility of reaching them effectively [34]. The first appearance of the concept of accessibility in the international literature was in the work of Hansen [35], who published an essay titled "How accessibility shapes land use". In this case, the measure of accessibility was used casually and routinely

for measures relating to the proximity of a place or people to all other places based on the notion of potential used in physics. Since this essay, many studies around the world have analyzed and applied various indicators of accessibility [36–40]. In most cases, these indicators were constructed to evaluate phenomena, such as innovations in the transport of people and goods [41], as well as the effect of distribution, equity, and levels of social exclusion [42,43].

The problem with the traditional principle of accessibility in urban economics is the limitation to measure, integrate, and analyze the socioeconomic and spatial reality in a more concrete way. Ramírez [33] stated that the shortcoming of this approach is its own validation, assuming the cost of eliminating from its analysis the role of subjects and social groups in the production and transformation of spaces. For Avellaneda [44], using this approach, all individuals were considered equal. On the other hand, from the perspective of daily mobility, the existence of a variety of subjects according to age, sex, social class, ethnic group, physical or mental condition, etc. is considered, each of which has different mobility needs—and accessibility—which is why various solutions are required. Cerda and Marmolejo [29] explained in this context that accessibility is not only determined by transportation networks, but it is also established by daily mobility as a physical reflection of people's will to move through the city.

## 2.2. Dimension of Accessibility in the Analysis of Sustainable Urban Forms

In addition, from accessibility, other determinants of sustainability can be discovered, such as the generation of needs for goods, services, and social contacts that give rise to the need to move, in which land use often appears as a determining factor [27]. In this context, it is argued that cities should be built so that access to opportunities is maximized without increasing the negative externalities that growth entails. According to the above, in a smart and sustainable transport framework, new urban structures should promote public transport, walking, and cycling, while discouraging private car use [23]. According to Kaufman and Widmer [45], accessibility is a mobility factor and is perceived as the provision of a service for its realization. (For example, the condition of sidewalks, bike lanes, public transport, networks, etc.). In this context, the existence of a poor service would indicate the existence of elements that would be functioning as "barriers", limiting access and preventing satisfactory mobility.

In this local context, within the matters related to accessibility perceived from a social point of view, Handy [46] stated that the residents of traditional neighborhoods, characterized by higher densities, with better accessibility and friendly design for pedestrians, showed more sustainable movement behavior than residents of neighborhoods with lower densities, poor accessibility, and unfriendly pedestrian design. In this context, streets are not only part of the urban infrastructure, but they are also organized according to the disposition people have when walking [12].

In this sense, pedestrian access should satisfy basic consumer needs, being a fundamental aspect of human activities [47,48]. In this context, accessibility, within the framework of a good sustainable urban form, is defined by the higher degree of satisfying the needs of residents who walk and use bicycles for commuting, in relation to the residents who drive automobiles [49].

## 2.3. Accessible Neighborhoods

To find out how sustainable a neighborhood is, according to the location of facilities that offer goods and services, there is a consensus in the literature stating that a facility that offers goods and services for basic consumer needs should be within 400 to 800 m from the households of residents [47–50]. Outside this limit, there would be a spatial barrier due to the distance. Specifically, the main consensus regarding the maximum distance residents who are willing to walk to facilities that supply goods and services in a residential neighborhood is 10 min on average [27,51,52]. In addition, according to Talen's ideas [24,53], the average speed of walking is 1.4 m/s (5.04 km/h) and it is classified as

"easy and healthy to walk around the neighborhood". In this case, a 5 min walk covers 400 m and a 10 min walk covers 800 m. According to this parameter, 400 m is the indicator to count how many residential land parts are within a radius of 400 m around a goods and services facility and to measure how sustainable a neighborhood under these criteria is [24–27]. Likewise, accessibility is also strongly conditioned by the state of sidewalks and lanes, as mentioned by citizens in the metropolitan area of the Greater Concepción [32], a matter to be considered in urban design.

### 2.4. Use of Geomarketing to Conduct Spatial Accessibility Analysis

One form of analyzing people's needs, specifically, the demand for goods and services, is through marketing [54,55]. Marketing related to spatial variables is known as geomarketing [56]. Geomarketing is widely used from a private perspective with the aim of monetizing sale businesses in facilities that offer goods and services [57,58]. This type of research has also been approached from a social perspective. Social marketing [59] focuses on the social well-being of the population through traditional marketing campaigns adapted to social factors. When applied to territory through urban spatial analysis, this strategy is social geomarketing [60].

According to this information, geomarketing is applicable to the private field [61–64] and social field, providing key spatial economic elements for project evaluation and the implementation of planning and urban design strategies, as stated by new available information technologies; specifically, the important contributions suggested by Escobar-Moreno [58] and complemented by Albornoz et al. [60].

Spatial Analysis: Delimitation of Market Areas

This type of analysis allows a higher knowledge of the potential market to be gained through the identification of the zone of influence [65,66], location of the competition, and establishment of optimal sale and service delivery points according to the spatial behavior of market data and social needs [67]. The delimitation of market areas can be prioritized by the proximity criteria, assuming that the most relevant factor is the spatial separation between supply (facility) and demand (households) points. In addition to this, besides evaluating the physical features of the territory, market areas also change with time due to the dynamism of the available supply, competition, or the potential demand [68]. This variability or change generally affects the spatial component [69]. Analyzing behaviors from a spatial–statistical point of view over time is fundamental when requiring projections for potential citizens/consumers and orienting sale strategies or urban design to analyze the spatial distribution of activities in relation to households.

This is oriented toward a microeconomic view that states how a society satisfies its needs, helping specialists to design specific plans according to demand. In this context, any type of campaign would be focused on social purposes, such as plans for sanitary assistance and urban design, among others [70–73]. Therefore, for the analysis of spatial demand, in general terms and considering consumer behavior [65], geomarketing makes the evaluation of the use of urban infrastructure and/or related services possible, allowing the creation of plans that can be integrated and movement analysis, considering the geographic location according to the type of industry or economic activity [74].

### 2.5. Geographic Information Systems as a Tool during a Sanitary Crisis

Geographic information systems (GIS) played an important role during the pandemic, making them a powerful tool to analyze the evolution of the situation during the sanitary emergency [75]. Together with medical and biological studies, which aim to find effective treatments for the disease, there are several contributions in the field of data analysis using GIS. The authorities' necessity to keep a track of confirmed cases favored the recollection of a large quantity of geospatial data, which have been subjects of study. In this way, the following analyses have been conducted: space–time analyses to predict virus spread [76], quarantine effectiveness [77], the correlation between the distribution of contagions and

social phenomena, such as migration during the pandemic's early stages [78], and the incorporation of environmental variables [79,80]. Diverse mapping web applications have used these same sources in order to generate real-time representations of pandemic evolution [81,82], with Dong Du and Gardner [83] being the most referenced. The presentation of data in a visual form has allowed an approach to present the information to the general public, in addition to working as a tool for researchers and health workers.

The effect of the pandemic on cities is also a common topic. Studies have covered themes related to access to certain health services during the pandemic [84,85] and the changes in dynamics within the city during lockdowns [19]. However, the roles that urban design and architecture have played as a support against the spread of the virus and during quarantines have not been considered relevant, and, many times, they have been absent from discussions about current and future contingency plans [86]. In some studies, the necessity to consider urban design in coming times is emphasized, that is, post-pandemic times and the creation of resilient cities when facing disasters of a similar nature [18,87].

The use of GIS in urban planning and for support in decision-making is not new. The use of GIS applications has been proposed for diverse tasks that involve selection between two or more options for the installation of infrastructures [88,89]. This tool has been used to assist the urban planning of cities in some regions. In Selamat et al. [90], a compilation of difficulties can be found, which were covered with the help of GIS in Malaysia.

## 3. Materials and Methods

### 3.1. Case Study: Description of the Experimental Area

The metropolitan area of Concepción (MAC) is a territorial unit made up of eleven communes in the Concepción Province, the eighth region of Chile. It is located in the central–south part of the country (latitude 36°35′ and 37°00′ S and longitude 72°45′ and 73°15′ W) and covers a total surface area of 2830.4 square kilometers, with 985,035 inhabitants, living mostly in urban areas [91]. It is considered the second-largest metropolitan area in Chile, according to demographic characteristics. The Concepción commune, a provincial and regional capital city, is the service center of the region and metropolitan area of Concepción. It has a total of 223,574 inhabitants and covers a surface of 221.6 square kilometers.

The testing area of the study is Collao, which is located in the northwest of the city, 2 km in road distance from downtown. A total of 20,383 inhabitants live there and represent 9% of the total city population. The Collao area is mainly a residential sector, with less than 3% of its space occupied for other purposes, such as goods and services, and green areas. Within its perimeter, some sites of communal interest can be found in the area, such as the Municipal Stadium, Chacabuco Military Regiment, Collao Bus Terminal, and University of Bio-Bio (Figure 1).

### 3.2. Data Used: Alphanumeric and Geographic Data

Alphanumeric data collection was carried out by conducting a random survey in a representative sample of the testing area according to social geomarketing techniques [60]. The questionnaire contained three differentiated sections: the first part collected data related to the socio-economic characteristics of the interviewees in order to segment mobility demand, the second part collected the relative preferences of movements to frequent activities, and the third part collected data related to the satisfaction level of the interviewees, such as the easiness and obstacles present in the movement.

The declared survey method was used to capture information about a population sample of individuals based on judgments declared by individuals about how they acted in different situations presented to them that must be as close to reality as possible. We worked with this type of survey due to its direct experimental nature with the preferences that the user declared. We interacted with the inhabitants of the Collao sector collecting information, in which the person declared the specific information for each section described above. Specifically, the survey asked about the socio-economic characteristics of the person, in order to explore the demand for segmented mobility, which made it possible to explore

different levels of accessibility and satisfaction. Consultations for activities, facilities, and barriers were characterized by daily activities carried out by people. Consultations for barriers were incorporated from the framework of accessibility from daily mobility [30] and from the dimensions of accessibility [32]. Finally, the section on preferences and capabilities included preferences for overcoming the identified barriers. The strategy responded to the citizen consultation methodology used to carry out the MAC mobility plan [92]. However, the inquiries made fell into the classification of intercept interviews prior to the pandemic, where the inhabitant was intercepted in the determined area with a concentration of activities. Given the nature of the study, during the pandemic, the telephone consultation was adjusted.

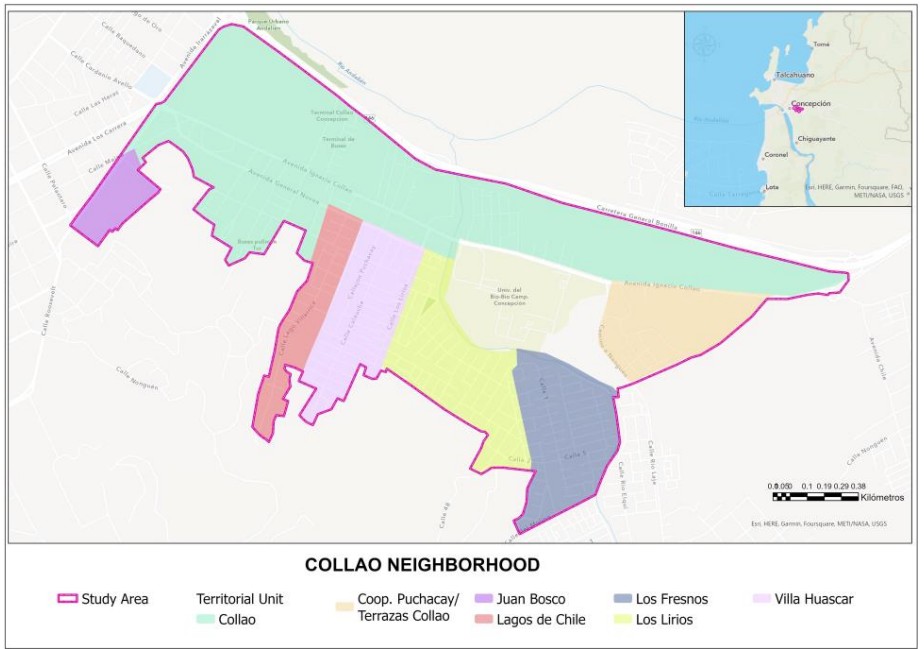

**Figure 1.** Case study area, Collao sector in the MAC. Source: Author's elaboration.

The designed questionnaires included three basic types of questions: closed response, graded response, and open response. A standard questionnaire was established for the residents of the study area, and the online format can be consulted at https://arcg.is/0Ojj4L0, which is also attached as annexed material. It was estimated that the sampling unit would be the inhabitants of the study area, who, at the time of being interviewed pre-pandemic, were traveling to carry out some activity with a destination facility in the Collao sector. During the pandemic, according to the confinement, the conversation took place in the context of the activities carried out during the use of health permits. The logic model of the survey is presented in Figure 2.

Data were collected in two stages: pre-pandemic, through perception surveys conducted in strategic places of the testing location, and during the pandemic, through phone surveys. During the first stage, 265 surveys were conducted. This number reached a total of 585 surveyed people added to the second stage. Given the total population (20,383), the sample is totally representative with a maximum of 4% error.

### 3.3. Geospatial Analysis: GIS Analysis through Geo Processes

Geospatial analysis has been particularly relevant due to the geographic impact of the pandemic and the number of space–temporal variables that need to be considered in order to achieve a valid interpretation of the phenomenon and decision-making that affect people's daily lives [93]. In this case, for data analysis, geo processes in ArcMap's® ModelBuilder were implemented and grouped into analysis models to evaluate the real accessibility and satisfaction levels, which are presented in Table 1.

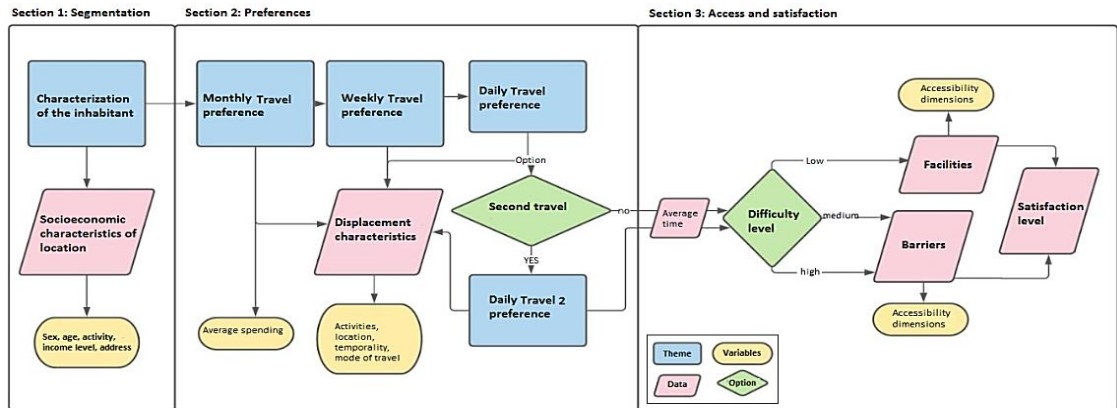

**Figure 2.** Logical model for collection and data analysis. Source: Author's elaboration.

**Table 1.** Model descriptions of the geoprocesses. Source: Author's elaboration.

| Geoprocess | Description | Input | Output |
|---|---|---|---|
| Accessibility PCBS Offer | Evaluates and delimits, in a real way, the spatial accessibility generated only by the infrastructure by type that offers goods and services of primary necessity used by the inhabitants. | Type of activity according to temporal frequency and analysis period (before or during the pandemic) | Map view, report files, and ArcGIS layers |
| Level Satisfaction | Estimates and delimits, in a real way, the satisfaction area according to the walk mode no more than 400 m from the facilities that offer goods and services in a territorial unit of analysis | Demanding households according to time frequency and analysis period (before or during the pandemic) that use the walk mode to places no more than 400 m away and declare a state of satisfaction | Map view, report files, ArcGIS layers |

### 3.4. Processes and Indicator Collection

The basics processes to carry out social geomarketing suggested by Albornoz, et al. [60] were used, adapted for the sanitary context produced by the COVID-19 pandemic. Within this work's conceptual framework, the analysis of accessibility through consumer preferences was conducted with inhabitants of the Collao sector during the COVID-19 pandemic in facilities where food and services are provided. The objective was to locate the facilities used by residents to stock up on food and acquire services with a temporal frequency focused on in the analysis through a survey that was previously applied and stored alphanumerically. When locating the facilities used, a real first indicator of pedestrian accessibility was calculated.

This indicator was obtained from the quantity of lots present within the area of delimited access, dividing them according to the total surface area of the study area. According to this, an average of lots that would have accessibility by the temporal frequency and studied typology was obtained. In this case, the main variables that allowed the calculation of this indicator could be considered, allowing the interface de spatial analysis to deliver a simulation of urban form in its accessibility dimension [24–26], and complement it with an analysis of the estimated satisfaction.

With the use of geomarketing, together with the obtainment of the aforementioned indicators, it was possible to spatially represent its delimitation, including the satisfaction detected through geoanalysis for the pre-pandemic period and during it. The geoanalysis of prior real accessibility, in this case, the spatial accessibility only possessed by the supply infrastructure used by the inhabitants, was evaluated in real form. This analysis

delivers two indicators that can be compared when conducting any spatial analysis: the real accessibility indicator and spatial coverage of pedestrian accessibility area.

To obtain the delimitation of areas of satisfaction, methodologically, the satisfaction level was specialized from the nearby category, as in all cases, it was the declared variable describing the satisfaction. Pre-pandemic, of the total number of respondents, 32% of the total number of records stated that they were satisfied with the proximity of the facilities that housed the activities carried out. In second place, their satisfaction would be justified by the mode of travel used, with 12.5%. During the pandemic, 35% declare closeness and 15% were satisfied with the health regulations applied. To represent the obtained levels, the same criteria that defined urban form in its accessibility dimension were used. Therefore, an area was delimited based on households that accessed facilities within no more than a 400 m walk and that also declared being satisfied. To locate and select the households, the records of the conducted survey that matched those characteristics were used.

## 4. Results

In the exploratory context of this research, the newly generated indicator was called "real accessibility", as the accessibility areas were delimited, including the preferences of the consumers of the Collao sector, based on the facilities actually used by residents monthly, weekly, and daily. The results obtained quantified the value of the resulting indicator. The new indicator was called "real accessibility" as it integrated the real preferences of use declared by the residents. The results are arranged into two categories. First, indicators of accessibility and its delimited spatial coverage are presented, obtained from the experience of movement declared by the people. Second, the areas of declared and delimited spatial satisfaction are presented according to the criteria of sustainability. This allows how satisfied the citizens are with the proximity to the dependencies that offer basic goods and services with no more than a 400 m walk to be spatially established.

### 4.1. Delimitation of Real Accessibility Based on the Preferences Declared by Citizens

Areas of accessibility were delimited, including consumer preferences, from the dependencies actually used by residents with monthly, weekly, and daily use according to the considered methodology. The indicator was referred to as "real accessibility" as it included real preferences declared by the residents of the Collao sector. Spatial results are presented through real coverage generated with the facilities used by the residents. Finally, a comparative synthesis of the data obtained before and during the pandemic was carried out, including cartography for the most representative temporal periods.

4.1.1. Real Accessibility to Goods and Services Pre-Pandemic

An initial comparative synthesis of the monthly, weekly, and daily periods for the indicator of real accessibility is presented. As previously mentioned, this indicator provides an initial approach to the real use of facilities offered by the Collao sector. According to this idea, Figure 3a shows the temporal frequency in which the facilities of the sector were mostly used based on the typologies of the activities analyzed. To analyze the values obtained, an ideal scenario was calculated for the Collao sector in which 100% of the territory was assumed to be covered under the influence within 400 m of all facilities for goods and services. In this ideal scenario, 100% of the batches would have had access to all facilities. Therefore, this value was obtained through the total number of lots in the study area, dividing it by the total area in hectares that the Collao sector covered. For this case, an accessibility value of 27.95 lots/hectare was obtained. According to this indicator, gaps could be estimated with the values obtained.

Given the value of ideal accessibility of 27.95 plots/hectare in the period prior to the pandemic, the temporal frequency closest to this number for the activities conducted was once or twice a week. For the service category, most of these processes were conducted once a month. In this way, prior to the pandemic, facilities for the provision of goods and services were used more frequently on a weekly basis. With a slight difference, facilities

with a monthly frequency were used. The coverage of real accessibility varied according to the analyzed temporal frequency.

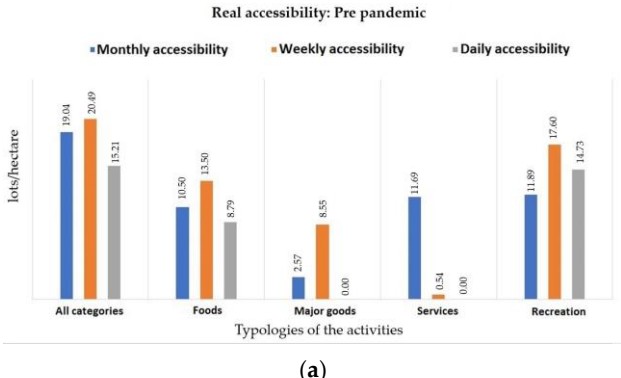

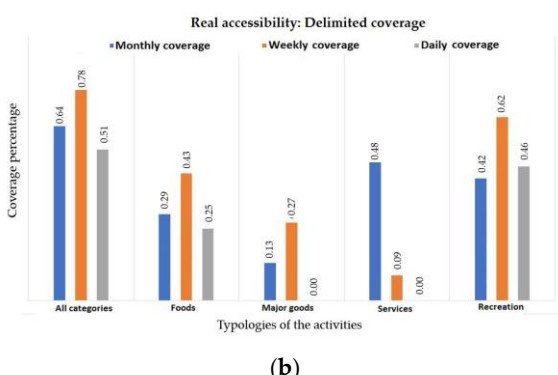

(**a**)

(**b**)

**Figure 3.** Indicators of accessibility per temporal frequency pre-pandemic. (**a**) Real accessibility. (**b**) Delimited accessibility coverage. Source: Author's elaboration.

To analyze the accessibility area, an ideal range would cover 100% of the study area. In the period prior to the pandemic, the accessibility coverage closest to this number was observed for the activities conducted once or twice a week. The weekly frequency included, in each category, a higher number of facilities near the inhabitant at a walking distance of 400 m, with the exception of the service category, as in its majority, these activities were conducted once a month. In this way, prior to the pandemic, goods and services facilities were used more frequently at a weekly level and would have had a wider coverage area. With a slight difference, facilities were used with a monthly frequency. Most of these activities were conducted in the study area (Figure 3b).

4.1.2. Real Accessibility to Goods and Services in the Pandemic

The same as prior to the pandemic, the weekly frequency included the highest indicators in each category. In this way, during the pandemic, the facilities providing goods and services were used with a higher frequency at a weekly level. According to the value of the indicators presented, the activities conducted with a daily frequency had a slight difference, also indicating that facilities in the Collao sector were used to a large extent, specifically those for activities related to the acquisition of basic goods, such as foods (Figure 4a).

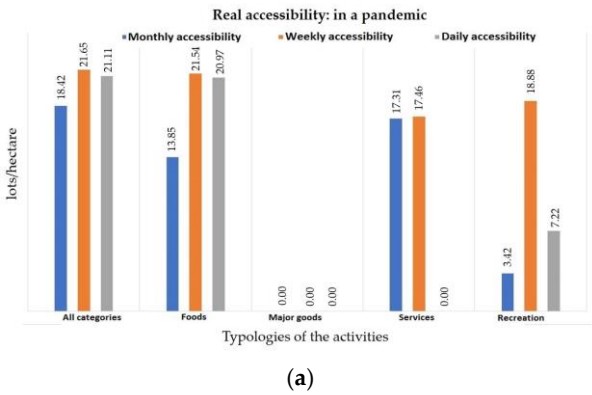

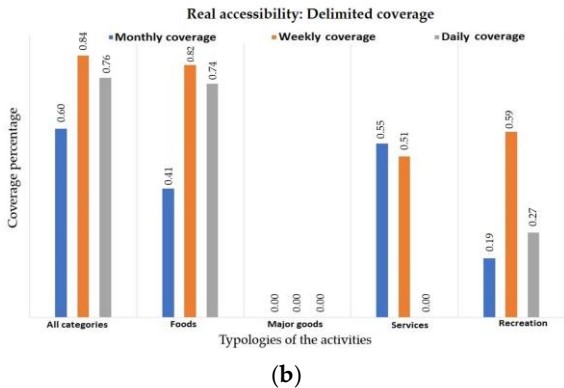

(**a**)

(**b**)

**Figure 4.** Indicators of accessibility per temporal frequency in the pandemic. (**a**) Real accessibility. (**b**) Delimited accessibility coverage. Source: Author's elaboration.

Based on the results presented previously and the accessibility indicators, the coverages of real accessibility also varied according to the temporal frequency analyzed (Figure 4b). Weekly frequency included, in each category, a higher number of facilities near the inhabitant at a walking distance of 400 m, with the exception of the service category. Most of these activities were conducted once a month.

This situation can be explained by the restrictions on the general mobility of people at a national level, as only two permits per week were given to engage in all kinds of activities. As a consequence, during the pandemic, facilities providing goods and services were used at a higher frequency weekly and exhibited a larger coverage area. With a slight difference, facilities were used with a daily frequency. This was proven by the necessity to acquire basic provisions at the shortest distance possible; therefore, the available facilities in the study area were used, which would exhibit good accessibility indicators.

### 4.1.3. Comparative Synthesis between Both Periods

There were differences between the levels of real accessibility and the resulting spatial coverages for the periods before the pandemic and during the pandemic. To present these variations, a comparison between the numeric and percentual differences was conducted. The graphs show the numeric decreases or increases in the indicator in a pandemic situation in relation to the normal situation (pre-pandemic). In the same manner, as seen in the results, when varying the real indicator of accessibility, the coverage percentage in the area of walking accessibility also varied in the same way in relation to the total area of the Collao sector. Figure 5a shows the numeric variation in the calculated indicator, and its delimited coverage is presented in Figure 5b.

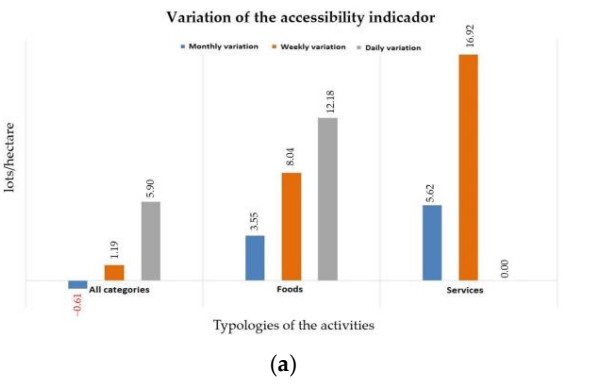
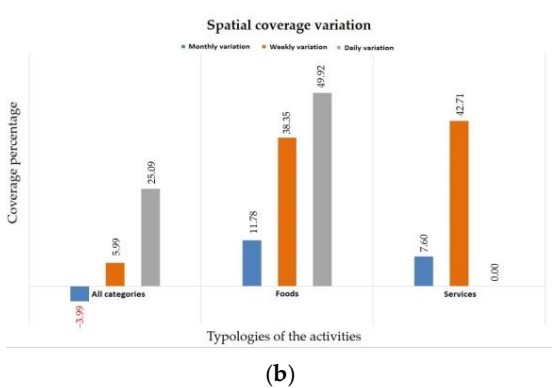

(**a**)        (**b**)

**Figure 5.** Variation in the calculated indicator and its delimited coverage. (**a**) Variation in real accessibility. (**b**) Spatial accessibility coverage variation. Source: Author's elaboration.

In Figure 5a, a wide variation in the real accessibility indicator for activities for acquiring foods during the pandemic can be observed. The accessibility of the available facilities increased their level in the monthly frequency. In the same way, the indicator showed a higher increment in its access level for the weekly and daily frequencies. This was also seen in the spatial coverage of the delimited area of accessibility (Figure 5b). This increment was due to the higher use of the facilities providing basic goods and needs in the Collao sector.

Two factors that conditioned the use frequency of the facilities had an effect in this case. The first was the number of weekly permits available to perform activities during quarantines. As there were only two, residents of the study area declared performing activities for their provisions as a priority at the moment of asking for a permit; therefore, the accessibility level of the Collao sector increased due to the wide offer of facilities providing food to residents. The second factor was the urban form of the Collao sector, which guaranteed the proximity of these facilities and their spatial distribution in the study area, also resulting in a great increment in the accessibility levels and its spatial daily coverage. As an example, the variation in the spatial coverage for the daily frequency is presented, as would have had a higher variation (Figure 6a).

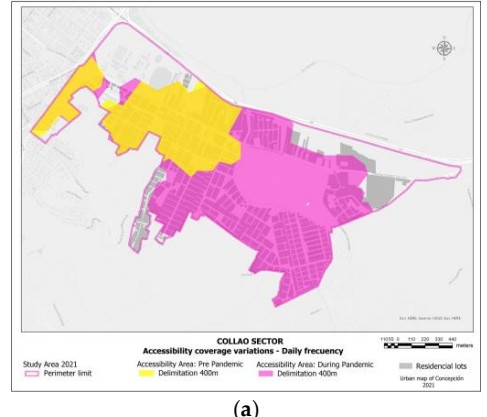
(**a**)

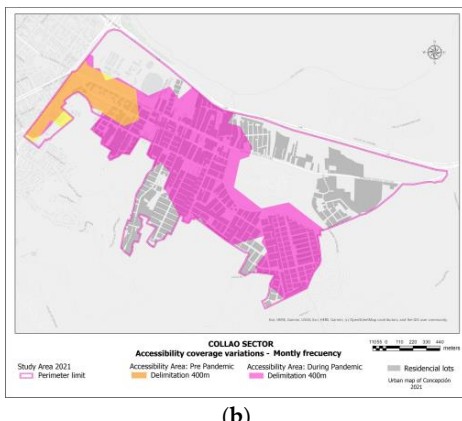
(**b**)

**Figure 6.** Comparison coverage of the delimitation of walking access to food and services. (**a**) Pre-pandemic spatial coverage. (**b**) Spatial coverage during the pandemic. Source: Author's elaboration.

Prior to the pandemic, carrying out these activities occurred around three nodes of high activity generated by large supermarkets within the study area. This assertion was based on the accessibility area generated from the used facilities, shown in yellow in Figure 6a. The described area would make a lower percentage of the Collao sector accessible, leaving the neighborhoods in the eastern part of the study area outside of the coverage. During the pandemic, the situation was different; all types of use of the facilities near the households of residents were restricted; therefore, the area provided higher accessibility, with a variation of 49.92% in relation to the period prior to the pandemic. It was confirmed that, in this case, by using less-preferred facilities before the pandemic, higher accessibility for residents was generated. In Figure 6a, this area is presented in violet.

Unlike the period prior to the pandemic, for the service typology, the level of accessibility and its coverage presented a significant positive variation, especially at a weekly level (+16.92 plots/hectares). This significant increment was explained by the realization of activities related to the acquisition of services, in its majority, within the study area. Given the mobility restrictions, residents of the Collao sector preferred using facilities destined for such purposes in the sector. According to this assertion, an increment in the coverage of the delimitation of walking access to these facilities was seen in comparison with the existent coverage prior to the pandemic. In this case, facilities providing services could receive higher use by the residents due to their dispersion in the space. This made it possible to walk shorter distances from the residents' households in the situation of a sanitary emergency. Figure 6b shows the differences in the spatial coverages at a weekly level. it can be seen that the accessibility in the period prior to the pandemic, represented in yellow, extended only in the vicinity of supermarkets, a clinic, and a bank. It is understood that such facilities were the most visited, generating an accessibility area of just 8.61% of the total of the Collao sector. This coverage extended over the residential neighborhoods of the Collao sector, covering in its total neighborhoods Los Fresnos, Los Lirios, and, to a larger extent, Villa Huascar and Lagos de Chile.

### 4.2. Delimitation of Areas of Spatial Satisfaction Declared by the Citizens

According to the levels of satisfaction prior to the pandemic, records in the surveys showed that 14.5% of the residents in the Collao sector declared that they were satisfied with the proximity of the activities they performed in a walking distance of no more than 400 m. The declared satisfaction corresponded only to the typology of foods. For the monthly frequency, the spatial coverage of satisfaction was 30%; the weekly frequency was 42%; and finally, the daily frequency was 26%. For the service typology, there was no coverage. In this form, the spatial satisfaction area for the weekly frequency is presented in Figure 7a, especially in the Collao sector. Additionally, it coincided with the analysis of accessibility for these activities.

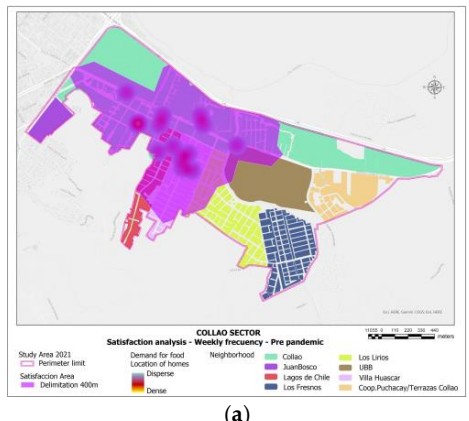

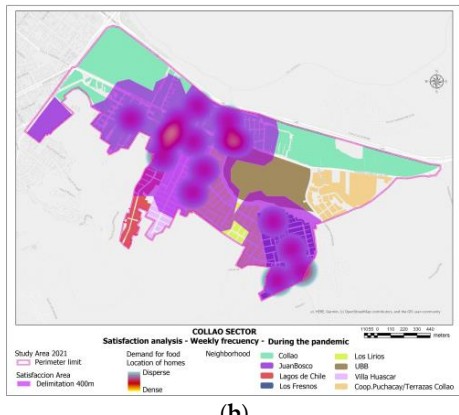

(**a**)                                                          (**b**)

**Figure 7.** Comparative synthesis between periods. (**a**) Pre-pandemic satisfaction. (**b**) Satisfaction during the pandemic. Source: Author's elaboration.

Regarding spatial satisfaction during the pandemic, the records in the surveys show that 28.7% of residents in the Collao sector declared that they were satisfied. The declared satisfaction corresponded to the typology of foods and services. The food typology, with a monthly frequency, had a spatial coverage of satisfaction of 38%, that of the weekly frequency had a coverage of 56%, and, finally, that of the daily frequency had a coverage of 55%. For the services typology with a monthly frequency, the spatial coverage of satisfaction is 14%, that of the weekly frequency was 19%, and, finally, there was no coverage for the daily frequency. The area of spatial satisfaction for the weekly frequency can be seen in Figure 7b.

The food category had a larger coverage of spatial satisfaction, especially at the weekly and daily levels. This coincides with the analysis of the accessibility to these activities. Services would have a certain degree of satisfaction due to the sanitary situation. In this context, this typology presented the activities in the study area that were conducted satisfactorily in the Collao sector. According to the sanitary context, the demand for foods at a daily level increased, which resulted in approximately 30% more in the spatial coverage of satisfaction. This increment in the coverage also indicated the increment in the use of the available facilities present in the Collao sector. Although all categories presented an increment in the satisfaction coverage, it is important to highlight that there was satisfaction, according to the increment in the demand for basic goods and services during the pandemic due to the sustainability in the urban form of the Collao sector.

### 4.3. Comparative Synthesis between Periods

According to the results, for the presented spatial satisfaction levels, there was a positive variation in the satisfaction coverage during the pandemic. During the pandemic, the delimitation of the coverage of spatial satisfaction for foods had a positive variation of 8%, which was 33% for a monthly frequency, 13.18% for a weekly frequency, and 29.17% for a daily frequency. For the service typology, there was a variation of 15.53% at the monthly and weekly levels, and 19.25% at the daily level. This indicates that, with a higher demand for goods and services within the Collao sector, the available facilities in the study area, according to its urban form, provided accessibility to residents, resulting in an increment in the coverage of spatial satisfaction declared by the inhabitants.

Finally, as an example, Figure 7a,b shows the geospatial variation in the delimitation of the satisfaction area. In Figure 7b, which represents the period of the pandemic, the resulting delimitation of geospatial satisfaction was 13.8% larger than that in the period prior to the pandemic represented in Figure 7a.

The increase in satisfaction was mainly described by the short-distance trips made, given the functional characteristics of the Collao sector to supply goods and services to its inhabitants. Specifically, of the 71.02% of the total people that declared satisfaction when

accessing the facilities during the pandemic, the variable that mainly explained access satisfaction was proximity to services, accounting for 48.85% of the total declarations. In second place was health regulations, with 21.26%. The third most-declared variable was the displacement mode, with 9.2%. However, the above takes on more force when relating these facilities to the declared displacement modal preference. In this case, it is evident that the walking mode concentrated the highest percentages of declared facilities, over 50%, for the three temporal frequencies analyzed. Second, the declared facilities were concentrated for motorized modes and, finally, cycling, as shown in Figure 8.

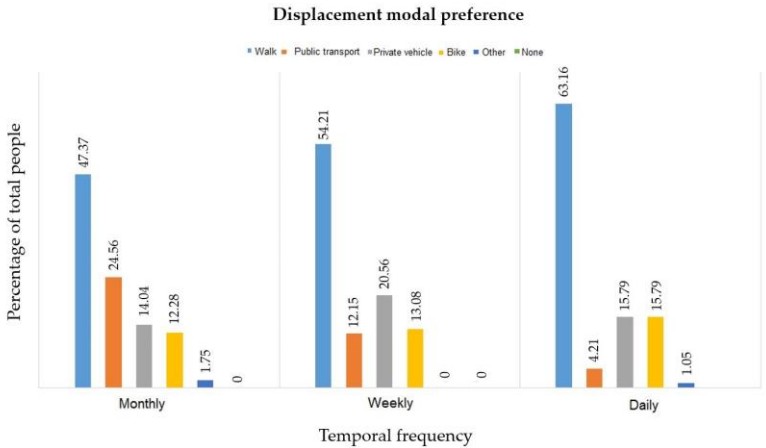

**Figure 8.** Displacement modal preference. Source: Author's elaboration.

This situation is explained by the large number of facilities that were described by proximity, health regulations, and the predominant mode used, given by the urban form that the Collao sector presented to generate accessibility to services.

## 5. Discussion

### 5.1. Contributions to the Current Knowledge

Table 2 shows a comparison between the spatial analysis presented and used in this article (COVID0359) and a sample of different GIS applications used in the research and problem resolutions related to the pandemic. The distinctive quality of the analysis is the possibility of conducting future simulations under different input parameter settings, which is innovative among similar applications. This same quality allows its use in urban planning and for the creation of public policies and control measures for post-pandemic times. The comparison criteria are based on those used by Franch-Pardo et al. [93,94].

In relation to the information used, this analysis deals with information obtained prior to and during the pandemic, which allows the consumer behavior regarding goods and services to be compared, a matter that was dealt with in Zecca et al. [19], but only on a daily basis and with visit times to the facilities. Other applications only refer to the diagnostics made with information collected during the pandemic. Additionally, this analysis was conducted on a local level, in comparison to others that were global. In relation to its utility for decision making, this research is a tool that allows both real cases in development and simulated situations to be analyzed, which was only previously conducted by Orea and Álvarez [75] and Zecca et al. [19]. Tools and techniques, in general, are common for all applications. In relation to the data topics, many applications were oriented to the detection of confirmed cases and their relation to other variables of pollution and infrastructure, among others. This research, just like that of Zecca et al. [19], was oriented toward the variables of accessibility and movement in real and simulated situations, with the difference that this research also considered trip objectives and orientated the results to urban design to contribute to social well-being.

**Table 2.** Comparative table of different GIS applications in the context of the pandemic.

| Analyzed Works | Geographic Area | Case of Use | Objective | Tools and Techniques | Period Analyzed | Data Topics |
|---|---|---|---|---|---|---|
| COVID0359 | Local, regionally scalable | Real case, possibility of simulating future scenarios | Support urban design based on accessibility and satisfaction indicators | Spatial statistics, network analyst | Pre-pandemic, during pandemic | Mobility, demographic, accessibility, satisfaction (survey) |
| Orea and Álvarez, 2021 [77] | National | Real case, simulated scenario | Analyze the effectiveness of the measures adopted to stop the spread of the epidemic | Spatial statistics, correlations | During pandemic, quarantine and non-quarantine | Confirmed cases, mobility |
| Rahman et al., 2021 [95] | Local | Real case | Examine the relationship between air quality indicators and disease transmission | Remote detection, correlations | During pandemic, quarantine and non-quarantine | Confirmed cases, contamination |
| Chen et al., 2020 [78] | Provincial | Real case | Determine the correlation between distribution of cases and emigration from Wuhan | Spatial statistics, correlations | During pandemic | Confirmed cases, suspected cases, deaths |
| Sajadi et al., 2020 [79] | Global | Real case | Examine association between climate and spread of the epidemic | Spatial statistics, correlations | During pandemic | Climate, confirmed cases |
| Dong Du and Gardner, 2020 [93] | Global | Real case | View and track cases | Web maps | During pandemic | Confirmed cases, deaths, people recovered |
| Zecca et al., 2020 [19] | Local | Real case, simulated scenario | Compare pedestrian flows and accessibility before and during epidemic | Network analysis | Pre-pandemic, during pandemic | Accessibility (survey), infrastructure, mobility |
| Shadeed and Alawna, 2021 [96] | Provincial | Real case | Determine which populations are most vulnerable in order to take appropriate actions | Multi criteria, risk map | During pandemic | Infrastructure, demographics, confirmed cases |

Note: Own elaboration based on the research cited Franch-Pardo et al. [93,94].

## 5.2. Indicator of Real Accessibility

The suggested modification involves a series of variations in the obtained results, therefore demonstrating the effectiveness of the used method, considering the main contributions that it makes to the evaluation of urban form in the accessibility dimension, a new indicator called "real accessibility". At a theoretical and methodological level, the main difference in the presented calculation is the possibility of conducting the calculation with temporal frequencies, an aspect that is not analyzed in experience [24–27]. According to this, the indicator of accessibility proposed by Talen [24] and that is widely used, does not allow it, providing only an approximation of the total disposition of the infrastructure providing goods and services to citizens. Unlike the above, in this research, the resulting value is quantified by including only the facilities that were actually used. The indicator was called "real accessibility" as, in this case, it only integrated the real preferences of use declared by the residents. Secondly, it allows the possibility of making a distinction

between the infrastructure actually used according to the identified temporal frequencies. In this case, when using fewer facilities than the total available in the study area, the accessibility indicator decreases. Pre-pandemic, in the food typology, the indicator of accessibility exhibited a negative variation of 14.63 plots/hectares at a monthly level. At a weekly level, the variation was −11.64 plots/hectares, and at the daily level, the variation was −16.35 plots/hectare. The same case is presented for the services typology, with a negative variation of −7.25 plots/hectare at a monthly level and 18.5 plots/hectare at weekly and daily levels. In the pandemic, in the food typology, the indicator of accessibility presented a negative variation of −11.29 plots/hectare at a monthly level. At a weekly level, a variation of −3.6 plots/hectare was observed and, at a daily level, a variation of −4.16 plots/hectare was observed. The same case was presented for the services typology with a negative variation of −1.63 plots/hectare at a monthly level and 14.9 plots/hectare at a weekly level.

The negative variation in the accessibility indicator was demonstrated, including the consumer preferences, in comparison with the indicator of relative accessibility used traditionally in the analysis of accessibility in the urban form. This situation is explained by the selective use of facilities per category. In this case, not all of the offers of available facilities are used. Differentiating between the offer of facilities that are used, among all the available offers, would allow a more precise spatial analysis of the accessibility coverage to be conducted according to the basic consumer needs. During the pandemic, the variation was lower as the accessibility coverages increased; hence, the real indicator was closer to the relative value.

When including these preferences and conducting the accessibility analysis only with the used facilities, the spatial coverage of the indicator decreased. When decreasing the number of facilities used to delimit the resulting accessibility area with a 400 m walk, in all cases, the spatial coverage decreased following the same tendency as the accessibility indicator. In this case, when including the consumer preferences and use of facilities, the spatial coverage responded only to the used facilities and not to all of the available offers. This adjustment confirmed the idea that the studied accessibility from a social perspective would allow adjusting strategies of urban design that would have an effect on the mobility policies aiming at sustainability and not unsustainable actions focused only on the resolution of traffic according to the ideas stated by Rode et al. [23].

*5.3. Indicator of Spatial Satisfaction*

Despite the obtained results being consistent with the initial ideas of Kaufman and Widmer [45], examining accessibility as a mobility factor and understanding it as a rendered service for its realization, the results contribute to our understanding of it as a basic characteristic in the urban form that a territorial unit has. This supports the idea of Dominguez [97], who stated that accessibility is a basic morphological characteristic understood as a condition that allows one to obtain, enter, feel, and use the diverse facilities offered by a city. In this way, the results allowed the evaluation of how satisfactory the movement to access a place is. This is related to the physical quality of the urban form, as explained by McCay et al. [98] and Zumelzu [26]. This made it possible to conduct an evaluation of the movement understood as an experience. Then, once the movement took place, the resident could be satisfied or unsatisfied with the urban services that made that movement possible.

According to this, when establishing an accessibility indicator, based on the evaluation of urban form in its accessibility dimension, the analyses of satisfaction were in this research frame. In this way, areas of relative and real spatial accessibility were evaluated, incorporating the consumers' preferences as a distance of no more than 400 m, a tolerable distance for walking. Consequently, to quantify their satisfaction, only residents who declared walking to their activities at no more than 400 m were selected. This argument allowed for evaluating the existing levels of satisfaction using measurements of accessibility in the same conceptual frame, aiming to determine the sustainability of the urban form.

Although the results for the delimitation of the satisfaction areas showed a low proportion in relation to the full sample, the number of many residents who declared being satisfied was established in a representative form according to the percentage of sample represented within the total. Consequently, as there was satisfaction and, according to Alonso and Grande [99], consumers tended to consolidate or improve their attitude to brands, reinforce their preferential scheme, and feel confident and predisposed to repeat purchases of the same brand, they will develop loyalty schemes.

Taking what was previously regarding the related satisfaction with a movement that allowed accessing a facility, the residents could consolidate their attitude toward the chosen movement form, reinforcing their preference and making them feel predisposed to repeat the experience. When using an indicator of an international standard to evaluate the sustainability in the urban form, how satisfied or unsatisfied the residents would be under this pedestrian standard was explored in this research, which was designed to evaluate the sustainability of the urban form present in a territorial unit. According to this, prior to the pandemic, the satisfaction percentage for the walk mode at no more than 400 m at a weekly level was 5.84%, being the highest in the period. During the pandemic, the percentage of the satisfied population was 10.5% at weekly and daily levels, and 6.12% at a monthly level.

These results allow for discussing the current satisfaction that a territorial unit provides in relation to the form in which a good urban form is established for the accessibility dimension. According to the obtained results, there would be a maximum of 10.61% of the population satisfied with their experience of real movement, walking a distance within 400 m. This value would show a breach to consider in order to increase the residents' satisfaction with the Collao sector, and the strategies of urban design implemented at the present time should be examined. This was discussed in the context of urban development, as stated by ODPM [100]; to develop sustainable neighborhoods, they must be safe, inclusive, well-planned in both construction and execution, and offer all residents the same access to opportunities, goods, and services. As the results indicated, the satisfaction was related to proximity, and there should be a degree to which residents experience strategies of sustainable urban design.

## 6. Conclusions

In relation to the research questions, it can be concluded that it was possible to incorporate and measure, through specific indicators, the preferences for movements declared by the citizens of the Collao sector in Concepción, Chile, thereby spatially delimiting the levels of satisfaction through social geomarketing techniques pre-pandemic and during the COVID-19 pandemic.

Based on the analysis, it is possible to change the location of infrastructure and vary the population, households, and land types to obtain indicators of accessibility according to real and relative parameters. In this way, it is possible to compare territorial units and simulate strategies of urban design to evaluate increases in declared special satisfaction. In this case, the application also allows for the analysis of data from standard surveys and compares temporal periods according to the values of the real preferences of residents in an area to recalculate and tune the indicators of accessibility according to the sustainable urban form.

These results demonstrate the importance of this type of analysis as a tool for supporting urban design decisions related to increasing green public spaces, non-motorized means of transport, and improving participatory and inclusive processes in the global south. Specifically, the tools presented in this research would allow for the reinforcement of relationships between the physical environment and the public sector in neighborhoods and other larger territorial areas at a regulatory level in the context of sustainable urban development and regional management post-COVID-19. It would also allow for better support for control and regulation decisions through urban regeneration programs, making the supervision, evaluation, and redesign of surroundings more efficient and promoting sustainable and safe access to goods and services. In short, this research proposes concep-

tual visions and a methodology to strengthen design decisions and implementation, and assess sustainable urban development processes in public spaces.

In addition, it can be considered as an instrument for supporting planning decisions and promoting sustainable urban development and regional management processes. This could allow for more objective responses to achieve initiatives associated with urban design to implement actions and strategies that consider social distancing in the short and long terms. In this sense, elements of the built environment, such as the width of sidewalks, front yards and trees, and other forms of street vegetation are positively perceived as emerging criteria for the healthy redesign of public spaces, being basic resources for social interaction and safe access to goods and services, especially during sanitary crisis times.

Recommendations for urban design in the post-COVID-19 period, in this type of context, must be focused on promoting the revitalization and improvement of the quality of public spaces, especially in terms of the quality and width of sidewalks and the urban image of public spaces, for example, providing local guides for street design. Sidewalks in neighborhoods and accessibility to public spaces must be guaranteed, and in their design, they must include vegetation elements in order to make them more attractive to pedestrians, in addition to taking care regarding vertical elements to avoid visual obstacles to guarantee social distancing and improve safety perception. Moreover, the improvement of spaces outside grocery stores, fruit shops, restaurants, and other services considering permanent spaces of at least two meters in width on sidewalks should be considered to increase the spatial satisfaction of citizens and foster loyalty to non-motorized transportation. It is important to include new means of regulation of urban design to achieve a specific quality for the urban form. At present, in Chile, current plans of regulation through zonation have been demonstrated to be sufficient, given the evidence of urban transformation and the insufficiency of regulatory management and control tools. In this sense, this type of application can be fundamental and necessary as new supporting tools for a higher precision for regulatory management and control in order to focus urban planning on more sustainable development. Future projects could use geoprocesses to estimate the geospatial demand level in order to determine and define spatial balance areas between the available area and spatial demand and explore its relationship with geospatial satisfaction.

**Author Contributions:** Conceptualization, F.N. and E.A.; methodology, E.A.; software, M.G. and E.A.; validation, F.N., A.Z. and E.A.; formal analysis, F.N. and E.A.; investigation, E.A. and F.N.; resources, F.N. and E.A.; data curation, E.A. and M.G.; writing—original draft preparation, E.A. and F.N.; writing—review and editing, E.A., F.N. and A.Z.; visualization, E.A. and M.G.; supervision, A.Z. and F.N.; project administration, F.N. and E.A.; funding acquisition, F.N. and E.A. All authors have read and agreed to the published version of the manuscript.

**Funding:** This research is a product of the finalized project "COVID0359". This project is titled "Safe accessibility to essential goods and services, in a confinement situation, as a new urban criterion in times of COVID-19, in the MAC, Chile". This project is funded by the Scientific Research Fund, Research Projects on COVID-19, and promoted by the National Research and Development Agency (ANID) of Chile.

**Institutional Review Board Statement:** Not applicable.

**Informed Consent Statement:** Not applicable.

**Data Availability Statement:** Not applicable.

**Acknowledgments:** The theoretical framework, methodological foundations, data, spatial analysis, and results, which the application calculated and delivered, are part of the doctoral thesis associated with the project COVID0359. The aforementioned thesis belongs to the doctoral program in Architecture and Urbanism at the Universidad del Bío-Bío, Chile. We are also grateful to our international collaborator, the Transportation Research Center (TRANSyT), belonging to the Universidad Politécnica de Madrid, Spain.

**Conflicts of Interest:** The authors declare no conflict of interest.

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
