# Peer review of "Socially Sustainable Accessibility to Goods and Services in the Metropolitan Area of Concepción, Chile, Post-COVID-19"

_sustainability, doi:10.3390/su142114042_

Round 1

Reviewer 1 Report

The Authors take as the main objective of this article was to explore and generate real accessibility indicators to goods and services, in addition to levels of spatial satisfaction of the population, at a regional level in the metropolitan area of Concepción, Chile. To focus on citizens’ social welfare, the Authors applied social geomarketing as method, obtaining the delimitation of accessibility areas for goods and services through population surveys and the delimited spatial decelerated satisfaction. Pre-pandemic and during-pandemic situations were evaluated. The results show an improvement in the delimitation of accessibility areas of goods and services since the citizens’ preferences as consumers are included, revealing an increment during the pandemic, especially the foods typology. In the same way, the existence of geospatial satisfaction and its increment in the pandemic context when accessing the diverse facilities that offer these kinds of goods is confirmed.  In my opinion, the paper can be published, after taking into account the following remarks:

- the paper title is too long now and should be shortened e.g. to "Socially sustainable accessibility to goods and services in the Metropolitan Area of Concepción, Chile in post-COVID-19",

- as far now, we can find in the Introduction section the background of the presented research topic as well as the scientific literature review. It would be good to prepare two separately sections: "Introduction" and "Literature review" for make this content more clear for readers,

- the subsection called "1.5. Case study: Description of the experimental area" should be included in the section called  "Materials and Methods",

- at the end of the Introduction section, the Authors should shortly write what was the main aim of the paper as well as what was contained in each paper section,

- In the Introduction section, the Authors wrote about many problems and issues connected with the covid pandemic. They wrote that in Chile, the covid pandemic has affected people’s mobility and access to urban activities. Yes, it is true, but it would be good to prove it by referring to the scientific literature in this regard, e.g. "Extracting road traffic volume in the city before and during covid-19 through video remote sensing", doi 10.3390 / rs13122329; "A Novel Method for Traffic Estimation and Air Quality Assessment in California", doi.org/10.3390/su14159169. One short paragraph in the Introduction section will be enough,

- the information presented in the figure called "Figure 1: Case study area, Collao sector in the MAC" is too small and poorly visible. Moreover, there is a lack of a legend. Moreover, the captions in figure 2 are too small...,

- in the subsection called "2.1. Data used: alphanumeric and geographic data", the Authors described the survey data used for further analysis. How aspects/questions were used? It should be a more detailed description of the used questionnaire provided,

- the paper text should be formatted according to the Sustainability journal paper template requirements. As far now, we can find some text without these requirements, e.g. tables, sots at the end of the sections/subsections names, etc.,

- subsection called "2.3. Processes and indicator collection" should not be divided into further sub-subsections, because their content is too poor, e.g. sub-subsection 2.3.1 consist only of two sentences! It is forbidden in serious scientific papers. The Authors could either develop the content of each sub-subsection or just not divide subsection 2.3 into further sub-subsections. This remark concern all similar cases in this paper,

-   "Figure 3: Indicators of accessibility per temporal frequency pre-pandemic": lack of names of the axis x and axis y, moreover the captions are too small and invisible. The same remark is dedicated to figure 4 and figure 5,

- figure 6, and figure 7: the legend is invisible.

Reviewer 2 Report

The authors attempt to investigate the levels of spatial satisfaction in the specific area of Collao before and during the COVID-19 pandemic. Although interesting may be the specific research topic, I fear that the article needs significantly more work before it could be published.

1.     First of all, the authors need to provide short conceptual definitions for the terms used. For instance, a short discussion of the notion of sustainable development should be provided in the introduction. See (a) Manolis Manioudis & Giorgos Meramveliotakis (2022) Broad strokes towards a grand theory in the analysis of sustainable development: a return to the classical political economy, New Political Economy, DOI: 10.1080/13563467.2022.2038114  and (b) Tomislav, K. (2018). The concept of sustainable development: From its beginning to the contemporary issues. Zagreb International Review of Economics & Business21(1), 67-94.

2.     The same also holds for the notions of urban sustainability and sustainable accessibility.

3.     The Introduction should be considerably reduced.

4.     In the abstract the authors claim that their aim is to “generate real accessibility indicators…” however I am not sure how this aim is finally accomplished. First the authors need to clarify what exactly do they mean by the term “real” and then explicitly highlight which are these indicators.

5.     Presenting their results, the authors argue that “the results show an improvement in the delimitation of accessibility areas of goods and services since the citizens’ preferences as consumers are included, revealing an increment during the pandemic, especially the foods typology. In the same way, the existence of geospatial satisfaction and its increment in the pandemic context when accessing the diverse facilities that offer these kinds of goods is confirmed. In conclusion, the satisfaction areas are useful for analyzing urban form designs and focusing them to promote the revitalization”. So, and if I understand correctly the level of citizens’ satisfaction has been changed during the pandemic period, however throughout the article no exegesis for this result has been provided. Given that the locations have not been changed in Collao during the pandemic, why residents’ satisfaction has been increased?

6.     In lines 490-491 what the authors mean by “allowing the comparison between the behavior of goods and services consumption?”. Arguably, the whole paper needs a comprehensive and detailed proof-reading.

7.     The 3.2.1. paragraph is repeated in lines 439-447.

Round 2

Reviewer 2 Report

Happy to see the authors accommodate many of my comments.

However, they should include the two references I have proposed.

See (a) Manolis Manioudis & Giorgos Meramveliotakis (2022) Broad strokes towards a grand theory in the analysis of sustainable development: a return to the classical political economy, New Political Economy, DOI: 10.1080/13563467.2022.2038114 and (b) Tomislav, K. (2018). The concept of sustainable development: From its beginning to the contemporary issues. Zagreb International Review of Economics & Business, 21(1), 67-94.

Round 3

Reviewer 2 Report

I would like to see this paper published